

# Investigation of aerosol indirect effects on monsoon clouds using ground-based measurements over a high-altitude site in Western Ghats

V. Anil Kumar, G. Pandithurai*, P. P. Leena, K. K. Dani, P. Murugavel, S. M. Sonbawne, R.D. Patil and R.S. Maheskumar

*Indian Institute of Tropical Meteorology, Pune - 411008, India*

*Correspondence to*: G. Pandithurai (pandit@tropmet.res.in)

**Abstract:** The effect of aerosols on cloud droplet number concentration and droplet effective radius are investigated from ground-based measurements over a high-altitude site where in clouds pass over the surface.
First aerosol indirect effect AIE estimates were made using i) relative changes in cloud droplet number concentration ($AIE_n$) and ii) relative changes in droplet effective radius ($AIE_s$) with relative changes in aerosol for different LWC values. AIE estimates from two different methods reveal that there is systematic overestimation in AIEn as compared to that of AIEs. Aerosol indirect effects (AIEn and AIEs) and Dispersion effect (DE) at different liquid water content (LWC) regimes ranging from 0.05 to 0.50 $gm^{-3}$ were estimated. The
analysis demonstrates that there is overestimation of $AIE_n$ as compared to $AIE_s$ which is mainly due to DE. Aerosol effects on spectral dispersion in droplet size distribution plays an important role in altering Twomey's cooling effect and thereby changes in climate. This study shows that the higher DE in the medium LWC regime which offsets the AIE by 30%.

**1. Introduction**

There exists a strong correlation between aerosol and cloud microphysics, more aerosols leads to many smaller cloud droplets for a fixed liquid water content, which increases the cloud optical thickness and cloud albedo [*Twomey*, 1947], this is termed as AIE which in turn cool the Earth's surface. But anthropogenic aerosols having
complex chemical composition will broaden cloud droplet spectral width and reduce the AIE [*Liu and Daum*, 2002]. Several theoretical and observational studies of aerosol-cloud interaction and estimation of AIE on maritime as well as continental clouds have been conducted worldwide. However, magnitude of this AIE still remains uncertain because of the complexity in the estimation [*Menon et al.*, 2008].





In the estimation of $AIE_n$ in many numerical models, the effect of dispersion is not considered, but the field studies of the indirect aerosol effect shows that polluted marine clouds and clouds of continental origin generally have both larger number concentration and a larger ε relative to clean marine clouds [*Liu and Daum*, 2002]. *Hudson and Yum* [1997] showed that the spectral width (σ) of cloud droplet spectra was greater with

higher $N_{ccn}$ from the Atlantic Stratocumulus Transition Experiment (ASTEX) conducted in 1992. They also noticed that this small σ difference can be noticed if cloud parcel with droplet of same mean diameter were considered. *Liu and Daum* [2002] shows the relation between relative dispersion of cloud droplet size distribution and number concentration of cloud droplets from 13 different experiments, of which 11 cases shows higher relative dispersion with higher $N_{ccn}$, and slight decrease in the other two.

*Pandithurai et al*. [2012] reported aerosol-cloud relationship and estimated AIE for warm continental cumuli over Indian sub-continent using Cloud Aerosol Interactions and Precipitation Enhancement Experiment (CAIPEEX) aircraft campaign data sets and observed influence of higher Cloud Condensation Nuclei (CCN) concentration on cloud droplet dispersion as it is significantly offsetting AIE over continental clouds. They also indicated that dispersion effect may be implicitly included in the estimation of AIEs derived using relative

changes in cloud effective radius ($R_{eff}$) with relative changes in aerosol concentration ($N_{acc}$). Also, *Hudson and Noble* [2013] argues that the increase in relative dispersion is largely due to decrease in droplet mean radius rather than increase in spectral width. However, *Liu et al*. [2013], *Pandithurai et al*. [2012] and the present study showed that the spectral width of droplet spectra shows increase if we consider a fixed mean droplet radius. *Martins et al*. [2009] showed that the increase in aerosol loading due to biomass burning aerosols

decrease the spectral width and in turn enhance the Twomey cooling effect. These contrasting reports suggest that more work is needed on this topic.

Thus in the present work aerosol effects on cloud microphysical properties from collocated simultaneous measurements over a high altitude station at Mahabaleshwar have been studied. Detailed analysis of AIE and its offset due to dispersion effect (DE) at different LWC regimes are presented.


## 2. Data Used

Recently set up High Altitude Cloud Physics Laboratory (HACPL) at Mahabaleshwar ($17.56^0$ N, $73.4^0$ E, 1348 m AMSL) has various ground based instruments for observation of aerosol as well as clouds. The observatory is situated in Western Ghats, wherein during monsoon season the temperature is varied between 17 and 23°C and

relative humidity varied between 85- 100% with an average rainfall of 500 cm. It is also observed that the site is covered by warm continental clouds most of the time during summer monsoon season. Interestingly the





observation from laboratory has shown that the total aerosol concentration in the size range (5 nm to 30 μm) varies from 100 to 25000 cm$^{-3}$ and occasionally it reached up to 40000 particles/cm$^3$ and CCN concentration from 400 to 20000 cm$^{-3}$. The aerosol and CCN concentration shows that the region is having higher aerosol concentration during monsoon season in spite of washout/scavenging due to precipitation (*Leena et al.,* 2015).

In addition to sea salt aerosols and local anthropogenic aerosol sources, there can also be secondary aerosols such as biogenic aerosols emitted from wet vegetation. In this work, we have made an attempt to study the influence of aerosol on cloud microphysical properties and their relationship using data collected during 2013 monsoon season. We have related cloud microphysical properties such as CDNC, R$_{eff}$ to aerosol concentration (N$_{acc}$) and CCN concentration (N$_{ccn}$) measured simultaneously by using various ground based instruments and

the details are described below. It is to be noted that for the present study data considered were explicitly for non rainy conditions. After scrutinizing the entire data set, about 20 hours of data were considered for analysis as detailed in Table 1.

### 2.1. CCN counter

The CCN counter from Droplet Measurement Technologies (DMT Inc.) was used for CCN concentration measurement. It was configured to operate at a fixed super saturation (SS) of 0.6% and sampling rate of 1 Hz. This CCN counter has a thermal gradient diffusion chamber in which super saturated water vapour condition is created. This supersaturated water vapour condenses on the CCN in the sample air to form droplets. An optical particle counter (OPC) using side scattering technology, counts and sizes the activated drops.


### 2.2. Cloud Combination Probe

Cloud microphysical parameters like cloud droplet number concentration, effective diameter and liquid water content were measured with Cloud Combination Probe (CCP) from DMT Inc, which is a combination of Cloud Droplet probe (CDP) to measure cloud droplet size distribution and concentration from 3 to 50 μm, categorized

into one of 30 channels. The CDP uses a laser to illuminate particles and forward-scatter their light. The detected scattered light is then used to size the particles. The intensity of the scattered light depends upon the size, composition, and shape of the particle. LWC was measured using a hotwire probe (HW-LWC) and also estimated from CDP drop size distribution measurements. For calculating spectral dispersion in cloud droplets and to estimate AIE, Cloud droplet mean radius (R$_m$), spectral width of DSD ($\sigma$) and relative dispersion ($\varepsilon$) were

calculated from the data obtained from CDP.





### 3. Analysis Procedure

As per Twomey's hypothesis, cloud droplet number concentration increases and cloud droplet size decreases with an increase in aerosol concentration for a fixed liquid water path (LWP). AIE can be estimated by utilizing both cloud droplet number ($AIE_n$) and droplet effective diameter ($AIE_s$) [*Feingold et al.*, 2003] through the following expressions:

$$AIE_n = \frac{1}{3}\frac{\Delta \log N_c}{\Delta \log \alpha} \quad ... (1) \qquad AIE_s = -\frac{\Delta \log R_{eff}}{\Delta \log \alpha} \quad ... (2)$$

where,

$\Delta N_c$ = the relative change in cloud droplet number concentration,

$R_{eff}$ = the cloud drop effective radius and

$\alpha$ = Aerosol or CCN concentration

$R_{eff}$ in the above relation is the ratio between third and second moment of CDP measured cloud droplet size distribution [*Hansen and Travis*, 1974]. The $R_{eff}$, is an important parameter in aerosol-cloud interaction studies, and can be expressed as

$$R_{eff} = \frac{\sum_{i=1}^{N} p_i r_i^3}{\sum_{i=1}^{N} p_i r_i^2} \quad ... (3)$$

where $N$ is the number of droplet sizing bins (30 bins for CDP), $p_i$ is the particles count for bin $i$ and $r_i$ is the mean radius in μm of bin $i$. For $R_{eff}$ parameterization, climate models use 1/3 power law which relates cloud liquid water content (LWC) and number concentration ($N_c$) [*Slingo*, 1990; *Bower et al.*, 1994; *Liu et al.*, 2006], and the same can be written as

$$R_{eff} = \beta \left( \frac{3L}{4\pi \rho_w N_c} \right)^{\frac{1}{3}} \quad ... (4)$$

where





$\rho_w$ = the water density

$\beta$ = effective radius ratio, which is a function of spectral shape of cloud droplet size distribution. Previous studies show that considering the dispersion effect is important for the AIE estimate (*Liu et al.*, 2008; *Peng and Lohmann*, 2013). However, this study uniquely different from other studies as it demonstrates that DE needs to be considered only in AIE estimates derived from relative changes in droplet number ($AIE_n$).

*Liu and Daum* [2000] and *Liu et al.* [2002] put forward an expression corresponding to Gamma distribution to describe the $\beta$ dependence on spectral droplet size distribution, which is as follows

$$\beta = \frac{\left(1+2\varepsilon^2\right)^{2/3}}{\left(1+\varepsilon^2\right)^{1/3}} \quad ... (5)$$

where '$\varepsilon$' is the relative dispersion of the droplet size distribution and can be defined as the ratio of standard deviation to the mean radius of the droplet size distribution. *Liu et al.* [2008] studied the relation between $\beta$ and ($L/N_c$) (water per droplet or specific cloud water content) to avoid uncertainties in the existing $\beta(\varepsilon)$-$N_c$ relation to explain dispersion effect. The relation between $\beta$ and ($L/N_c$) can be expressed as

$$\beta = \alpha_\beta \left(\frac{L}{N_c}\right)^{-b_\beta} \quad ... (6)$$

where $b_\beta$ is the dispersion factor defined as the percentage of offset/enhancement due to Twomey cooling effect, due to the dispersion occurred in the cloud droplet size distribution.

*Shao and Liu* (2006) proposed that the real strength of the AIE should be sum of the Twomey's cooling effect and the compensating effects such as droplet dispersion effect and influence of entrainment mixing.

## 4. Results and Discussion

To understand the interaction of aerosol with cloud microphysical parameters (CDNC and ED), aerosol and CCN concentration for a fixed LWC (0.15 gm$^{-3}$, for e.g.) were plotted and displayed in Figure 1a and 1b. It can be noted from these figures that, higher aerosol/CCN causes more number of cloud droplets (Fig 1a). Higher droplet regime is having a smaller effective droplet diameter. For estimating AIE, CCN concentration, cloud droplet number concentration and droplet effective diameter at different liquid water contents were examined. The relationship between aerosol-CCN-CDNC and aerosol-CCN-Droplet effective diameter for three different LWC bins was shown for better clarity to the readers. Figure 2 depicts the relation between the cloud





microphysical parameters, CDNC, Droplet Effective Diameter (ED) and LWC. It clearly shows the reduction in the ED with increase in CDNC for all LWC bins. Sharing the same available liquid water content by more cloud droplets leads to reduction in its size.

**4.1. AIE estimation from droplet effective diameter and number concentration**

Figure 3a shows the variation of CDNC with CCN concentration for three different narrow LWC bins of 0.20, 0.21 and 0.22 $gm^{-3}$. It can be noted that CDNC is increasing with increasing CCN concentration. It is seen that they are well correlated (statistically significant) with a correlation coefficient of 0.68, 0.63 and 0.62, respectively for the above three LWC bins. The value exponent of power law fit between CDNC and CCN were

0.24, 0.23 and 0.22. This results into $AIE_n$ estimates of 0.08 and 0.076 and 0.074 respectively.

We have also seen the variation of $R_{eff}$ with CCN concentration (Figure 3b) and it is seen that the reduction in $R_{eff}$ with increase in CCN concentration. The linear fit to log-log plot of $R_{eff}$ and CCN shows a correlation of -0.55, -0.51 and -0.50 for individual LWC bins where the value of exponent ($AIE_s$) varies as -0.055, -0.051 and -0.050, respectively for LWC bins of 0.20, 0.21 and 0.22 gm-3. From this, it is clear that increased CCN

concentration led to more number of cloud droplets of smaller size, which is generally termed as aerosol indirect effect, represented herein by $AIE_s$. It can be noted that $AIE_n$ values are higher about 30-40% as compared to $AIE_s$. This clearly suggests that the overestimate of $AIE_n$ may be due to some other effects such as dispersion and entrainment. Some of the previous investigations of $AIE_n$ using aircraft measurements generally relate sub-cloud aerosol measurements with in-cloud droplet number concentrations without grouping it into constant

LWC bins and attributed the overestimation to entrainment mixing processes (*Shao and Liu*, 1996).

Further, *Shao and Liu* (2006) argued that the systematic discrepancy between $AIE_n$ and $AIE_s$ is caused primarily by the differential loss of cloud liquid water between clean and polluted clouds. Previous studies have also shown that droplet dispersion effect can partly offset the Twomey's cooling effect. However, there is no clear demonstration of observed differences between $AIE_n$ and $AIE_s$. This warrants the study why $AIE_n$

overestimate $AIE_s$. This study clearly provides an observational demonstration of overestimation of $AIE_n$ as compared to AIEs and the droplet dispersion effect offset brings down the $AIE_n$ close to $AIE_s$.

Before attempting to estimate the aerosol indirect effect, we have analyzed the LWC variation of non-precipitating clouds for the period considered in this analysis and presented in Figure 4 as frequency distribution of LWC. It can be seen that it is varied from 0-1 $gm^{-3}$ with a maximum percentage between 0.05 to 0.5 $gm^{-3}$.

Thus considering this, aerosol indirect effect in terms of number concentration and droplet size has been



calculated for each bin of liquid water content. From Figure 5, it is observed that beyond $0.35 \, gm^{-3}$ of LWC, the variability of the values seems to be high due to less availability of data and lower statistical significance which is separated by a dotted line in the figure. For present analysis of AIE, CCN and cloud microphysical properties (CDNC, Deff) were first grouped by LWC for which a bin size of $0.01 \, gm^{-3}$ LWC was selected. In Figure 5, the

aerosol indirect effect and dispersion factor for the mentioned LWC ranges is represented, in which green line corresponds to $AIE_s$, black line corresponds to $AIE_n$ and blue line shows dispersion factor.

The estimated AIE shows its maximum value at lower LWC regime, reduced to half in the moderate LWC regime and falls to minimum at higher LWC range. In this observation, the maximum AIE estimated from droplet concentration and size is 0.103 ($AIE_n$) and 0.088 ($AIE_s$), respectively. *Pandithurai et al*. [2012] using

CAIPEEX aircraft data, reported AIE over Indian region from both methods are found to be 0.13 ($AIE_n$) and 0.07 ($AIE_s$). In the lower LWC regime, reduced collision coalescence may cause large number of smaller droplets with almost same size, which increases the AIE and reducing the dispersion factor.

**4.2. Dispersion Effect**

It is well known from earlier studies that droplet relative dispersion ($\varepsilon$) is an important cloud microphysical

parameter which is defined as the ratio of spectral width of cloud droplet size distribution ($\sigma$) to mean radius ($R_m$). *Liu and Daum* [2002] noted that higher CCN concentration will cause broader cloud droplet spectra, which will reduce cloud albedo thus tend to reduce the AIE. From Figure 6a, it can be seen that the relative dispersion is increasing with CCN concentration which is similar to the previous studies [*Liu and Daum*, 2002; *Pandithurai et al*., 2012].

Apart from this, Figure 6b shows the observed variation of $\sigma$ with CCN concentration, for fixed mean radius for three narrow ranges 4 - 4.5µm, 4.5 - 5µm and 5 - 5.5µm, in which $\sigma$ is increasing with $N_{ccn}$. This observation is similar to that reported by previous investigators through aircraft measurements [*Hudson and Yum*, 1997; *Miles et al*., 2000; *Pandithurai et al*., 2012]. *Hudson and Yum* [1997] demonstrated that standard deviation of cloud droplet spectra will be higher at higher CCN concentration from the ASTEX experiment of 1992. In general,

clean maritime clouds have larger droplet size and larger $\sigma$, while polluted continental clouds have smaller drops with smaller $\sigma$. According to *Yum and Hudson* [2005], the large dispersion observed in the continental polluted cloud was possibly due to smaller droplets rather than broader droplet spectrum. From this observation, we can see that the increase in relative dispersion in polluted continental clouds is not only because of the reduction in droplet radius at higher $N_{ccn}$ but also due to increase in $\sigma$. According to *Liu and Daum* [2002], anthropogenic





aerosols have a complex chemical composition which leads to broader activated drop size distribution, and those small droplets in the cloud compete for water vapor and broaden the size spectrum.

Further, effective radius ratio, $\beta$, which is an increasing function of relative dispersion, $\varepsilon$ of cloud droplet spectra, has been calculated from CDP data. The value of $\beta$ varied from 1.06 to 1.28. Similarly the value of $\varepsilon$ varied from 0.27 to 0.59. Higher $\varepsilon$ corresponds to higher CCN concentration. Figure 7 shows the relation between water per droplet (L/N) and effective radius ratio, $\beta$. It is found that $\beta$ decreases with L/N and the linear fit to the data gives the following relation with a correlation coefficient of 0.73. The dispersion factor estimated from whole data set is 0.07. A value of $b_\beta = 0.07$ indicates that the dispersion can offset AIE by 21%.

**4.3. AIE estimation considering dispersion effect.**

Earlier studies [*Peng and Lohmann*, 2003; *Liu et al*., 2008 and *Liu and Daum*, 2002] have reported that the dispersion effect tends to offset the AIE in the range from 10 to 80%. On the contrary, *Martins et al*. [2009] reported that the dispersion effect enhances the Twomey cooling for biomass burning aerosols in the dry region. From the present study, the maximum dispersion factor $b_\beta$ found to be 0.098 for LWC of $0.21 gm^{-3}$, the corresponding $AIE_n$ was 0.077, the dispersion effect is found to be -0.023, which can offset the $AIE_n$ effect by 29.6%, so the resultant $AIE_n$ is 0.054 (0.077 – 0.023). This resultant $AIE_n$ is close to the estimated $AIE_s$ (0.051) using $R_{eff}$ for the same LWC bin. From this, we can clearly state that DE is implicitly included in the estimation of $AIE_s$. This analysis is presented in Figure 8 which shows the variation of $AIE_s$ and resultant $AIE_n$, i.e., $AIE_n$-DE for different LWC regimes along with dispersion offset in percentage. DE is very less at lower LWC regime where AIE has its maximum value; DE tends to increase with LWC and shows maximum at moderate LWC regime then found to be reducing at higher LWC regime. From the figure we can see that resultant $AIE_n$ is closely agreeing with $AIE_s$ for all LWC range except in the lower LWC where dispersion effect is too small, and the AIE has its maximum value.

**5. Summary and Conclusion**

Aerosol indirect and dispersion effects on monsoon clouds over a high altitude site in Western Ghats using in-situ instruments were studied. Advantage of simultaneous measurement of aerosol and cloud microphysical properties have been utilized from this observatory and studied i) discrepancy between aerosol indirect effect estimates from different methods, ii) the variation in albedo susceptibility with LWC, iii) the dispersion effect variation with LWC and the resultant AIE. Over Indian subcontinent, the relative dispersion found to increase



with aerosol concentration in continental clouds sampled through aircraft measurements over several regions during the CAIPEEX experiment [*Pandithurai et al.,* 2012; *Prabha et al.,* 2012]. Higher concentration of anthropogenic aerosols with different chemical composition and sources can contribute to cloud droplets of different sizes. It was also observed that an increase in spectral width of the drop size distribution ($\sigma$) with $N_{ccn}$

for a given mean cloud droplet radius, indicating a broader spectrum at high aerosol concentration. As *Liu* [2002] suggested, this increasing tendency of $\sigma$ and hence $\varepsilon$ might be because of several possible reasons, such as mixing of clouds at different developmental stages, or may be due to strong collision-coalescence.

The AIE found to be maximum in low liquid water content clouds but at the same time dispersion of cloud droplet size distribution is very less. While considering clouds with moderate liquid water content we can see an

increase in dispersion effect which offsets the AIE. At higher LWC, the dispersion as well as AIE seems to be reducing. In this study, we investigated the effect of aerosol on cloud effective radius and droplet number concentration. The estimated aerosol indirect effect derived from effective radius changes (i.e., $AIE_s$), the maximum observed is 0.088, and effect on droplet concentration, $AIE_n$, the maximum found is 0.103, similarly the maximum dispersion offset obtained is 29.6%. From the resultant $AIE_n$, it is clear that in AIE estimate using

droplet concentration, dispersion effect should be taken in to account for correct estimate of AIE. It is also confirmed that AIE estimation from effective droplet radius implicitly includes DE. As the droplet effective radius is a ratio of third and second moment of DSD, the dispersion in DSD is implicitly included and hence the AIE estimates obtained through relative changes in observed $R_{eff}$ does not need to consider the dispersion effect.

**Acknowledgements**

Authors are grateful to all the team members of High Altitude Cloud Physics Laboratory (HACPL), of IITM. HACPL is fully funded by Ministry of Earth Sciences (MoES), Govt of India, New Delhi. The data used in this study is from the data repository of HACPL, part of IITM, Pune and the data can be made available on request with a due approval of Director, IITM.  Authors are thankful to the Director, IITM for his support and

encouragement.






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





**Table 1**. Data sampling dates/periods used in this study during which no precipitation was recorded.

| Date | From | To | Total (minutes) |
|---|---|---|---|
| 06/06/2013 | 16:07 | 16:40 | 33 |
| 11/06/2013 | 8:38 | 9:13 | 35 |
| 12/06/2013 | 9:10 | 12:37 | 207 |
| 23/06/2013 | 9:54 | 10:59 | 65 |
| 15/07/2013 | 22:35 | 22:51 | 16 |
| 25/07/2013 | 9:53 | 10:37 | 41 |
| 07/08/2013 | 19:51 | 22:40 | 169 |
| 16/08/2013 | 10:04 | 17:19 | 75 |
| 18/08/2013 | 18:02 21:00 | 20:39 23:02 | 279 |
| 19/08/2013 | 19:58 | 23:01 | 183 |
| 20/08/2013 | 22:04 | 23:31 | 87 |
| **TOTAL** | | | 1190 |



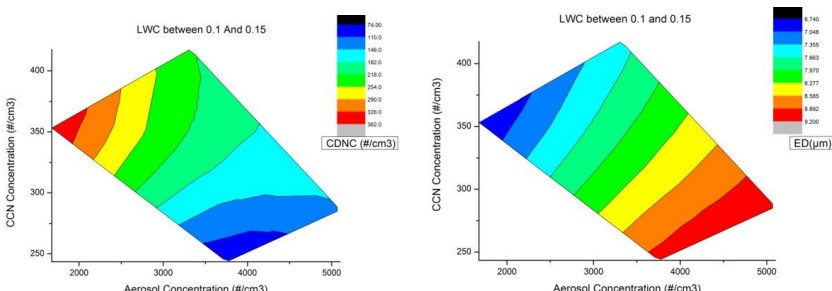

5  **Figure 1 (a).** Covariation of aerosol-CCN-cloud droplet number as observed at a high-altitude site for the LWC range 0.1-0.15 gm$^{-3}$ relation of Aerosol, CCN , CDNC, **(b)** same as (a) but aerosol-CCN-cloud droplet effective diameter.

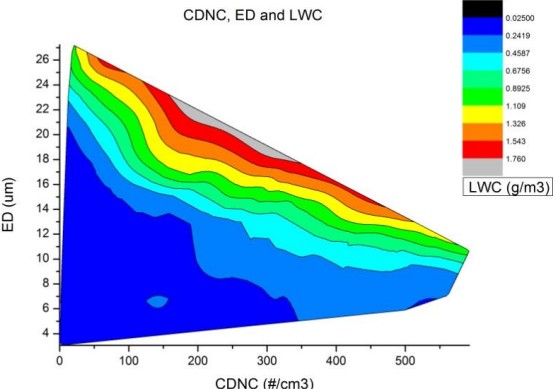

**Figure 2.** The relationship between cloud droplet number concentration (CDNC), droplet effective diameter
10  (ED) for different LWC values. CDNC and ED showing an inverse relationship for different LWC bins.



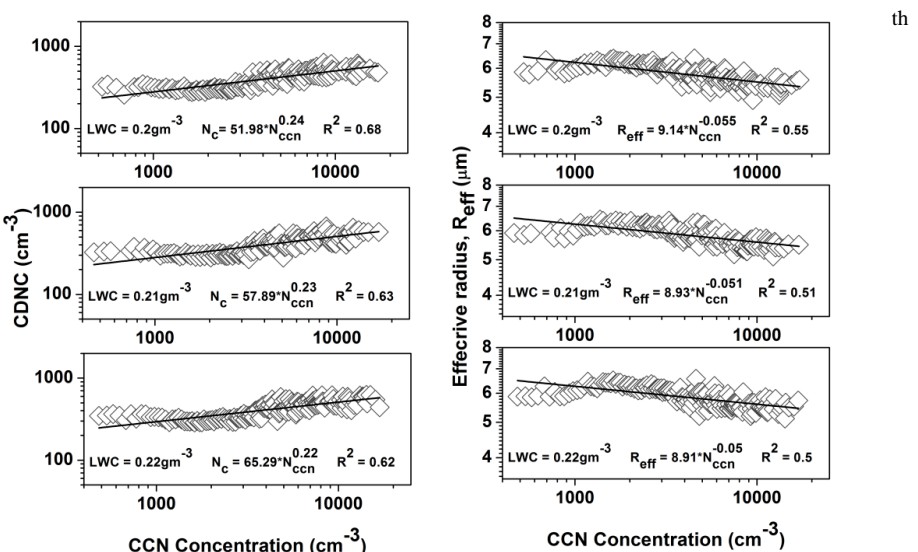

**Figure 3 a and b**. Variation of CDNC and effective radius with CCN concentration at 0.6% supersaturation for
15   three LWC bins namely 0.20, 0.21 and 0.22 gm⁻³.

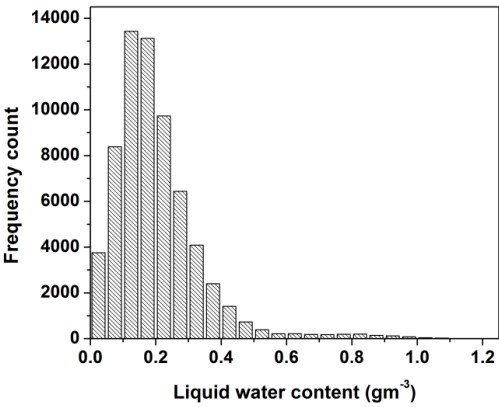

**Figure 4**. Frequency distribution of LWC data considered in the present study





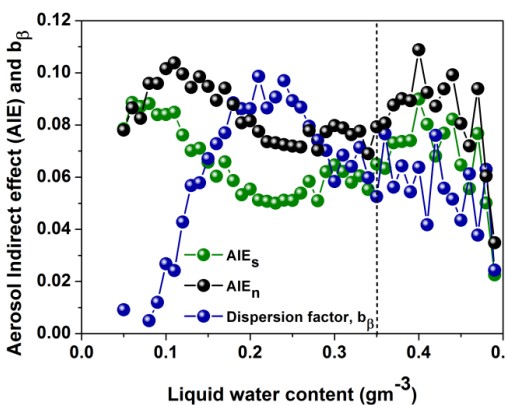

**Figure 5**. Aerosol indirect effect estimates using CDNC (AIEn), $R_{eff}$ (AIEs) and dispersion factor for the LWC ranges is represented. Green line corresponds to AIE$_s$, black line corresponds to AIE$_n$ and blue line shows dispersion factor. AIEn is higher than AIEs.

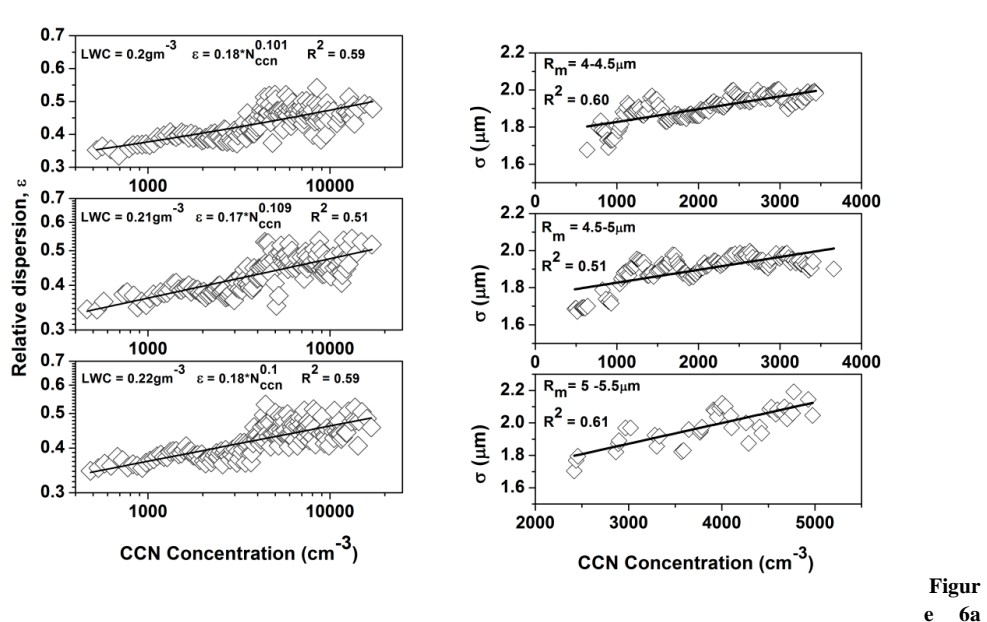

**Figure 6a** and **6b**. Change in relative dispersion and spectral width of cloud droplet spectra with CCN concentration.





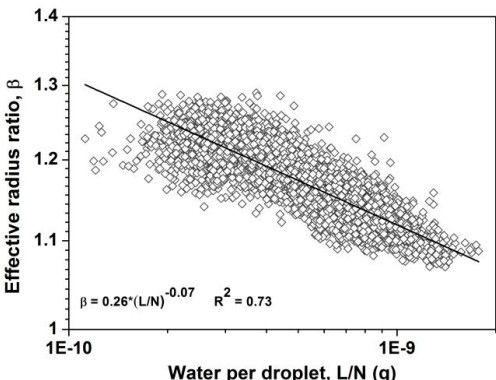

**Figure 7**. Relation between effective radius ratio and water per droplet

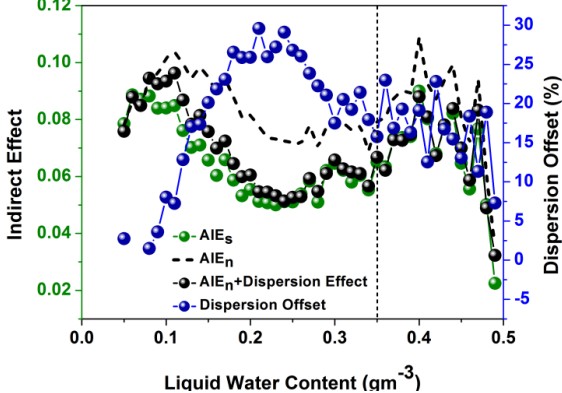

**Figure 8**. Aerosol indirect effect estimated after considering dispersion effect in $AIE_n$, Left axis represents AIE,
right axis represents dispersion offset in percentage.