# Peer review of "Investigation of aerosol indirect effects on monsoon clouds using ground-based measurements over a high-altitude site in Western Ghats"

_Atmospheric Chemistry and Physics, 2015_

## Referee Comment (RC1) · Anonymous Referee #1 · 29 Feb 2016

Overall summary:

This manuscript used ground-based measurements at a high-altitude site over Western Ghats to estimate the aerosol effect on cloud droplet number concentration (AIEn) and droplet effective radius (AIEs). For a given LWC, they found the AIEn is larger than AIEs, which could be explained by the cloud droplet dispersion effect. And then they finally demonstrated that. This work presents valuable information to compare or calculate the AIE from different methods. Some minor questions/suggestions need to be solved are listed in the following:

Comment and Question:

[Figure]

1. Page 2, Line 3: Authors should define the parameter (herein, $\varepsilon$) or acronyms when it firstly appeared in the article.

2. Page 2, Line 5: 'They also noticed that this small difference can be noticed ' This sentence should be modified.

3. Page2, Line 8: 'shows' should be changed to 'show'

4. Page2, Line 9: 'slight' should be changed to 'slightly'. There are English editorial issues throughout the paper. Authors should pay more attention on it.

5. Page 2, Line 30: The unit of average rainfall should be 'cm/yr'.

6. Authors should give a brief introduction about sources and types of aerosols in this site. And the size range of aerosol instrument, ranging from 5nm to 30$\mu$m, is very broad. Details about quality of surface instruments and calibration should be added.

7. Define of $\alpha\beta$ in equation 6

8. For Figure 1 and 2, the LWC is fixed about 0.15g/m-3, while in Figure 3 the LWC is fixed from 0.20g/m-3 to 0.22g/m-3. Why don't use the consistent LWC bin?

9. .Page 8, Line 9: explain the calculation step from 0.07 to 21

10. Cloud effective radius going down with aerosol concentration increasing might be due to the aerosol indirect effect as the author discussed in this article. It might also be due to the aerosol semi-direct effect (Huang et al., 2006a, 2006b, 2008 and 2010, 2014), which is very popular over the Asia region. Author should clarify this problem in the article.

Huang, J., P. Minnis, B. Lin, T. Wang, Y. Yi, Y. Hu, S. Sun-Mack, and K. Ayers, Possible influences of Asian dust aerosols on cloud properties and radiative forcing observed from MODIS and CERES, Geophysical Research Letters, 33 (6) (2006a), L06824, doi:10.1029/2005GL024724.

Huang, J., B. Lin, P. Minnis, T. Wang, X. Wang, Y. Hu, Y. Yi, and J. Ayers, Satellite-based assessment of possible dust aerosols semi-direct effect on cloud water path over East Asia, Geophysical Research Letters, 33 (19) (2006b), L19802, doi:10.1029/2006GL026561.

Huang, J., P. Minnis, B. Chen, Z. Huang, Z. Liu, Q. Zhao, Y. Yi, and J. Ayers, Long-range transport and vertical structure of Asian dust from CALIPSO and surface measurements during PACDEX, Journal of Geophysical Research, 113 (D23) (2008), D23212, doi:10.1029/2008JD010620.

Huang, J., P. Minnis, Yan, H., Yi, Y., Chen, B., Zhang, L., and J. K. Ayers, Dust aerosol effect on semi-arid climate over Northwest China detected from A-Train satellite measurements, Atmos. Chem. Phys., 10 (2010), 6863-6872.

Huang, J., T. Wang, W. Wang, Z. Li, and H. Yan, Climate effects of dust aerosols over East Asian arid and semiarid regions, Journal of Geophysical Research: Atmospheres, 119 (2014), 11398–11416, doi:10.1002/2014JD021796.
* * *

---

## Author Comment (AC1) · 30 Mar 2016

Response to the Anonymous referee #1

We would like to thank the Anonymous referee#1 for his valuable suggestions which helped in improving the quality of this paper. Suggested corrections and additions were incorporated in the revised manuscript.

Q1: Page 2, Line 3: Authors should define the parameter (herein, $\varepsilon$) or acronyms when it firstly appeared in the article.

Reply: Thanks for the suggestion and accordingly the parameter ÉŻ is defined on its first occurrence (Pg #2 Line #3) in the revised manuscript.

Q2: Page 2, Line 5: 'They also noticed that this small difference can be noticed'. This sentence should be modified.

Reply: Corrected. (Page #2, Line #6) The above sentence is rewritten as follows: "They also observed that this small $\sigma$ difference can be noticed if cloud parcels with droplets of same mean diameter were considered".

Q3: Page2, Line 8: 'shows' should be changed to 'show'

Reply: Corrected. (Page #2, Line #9)

Q4: Page2, Line 9: 'slight' should be changed to 'slightly'. There are English editorial issues throughout the paper. Authors should pay more attention on it.

Reply: Corrected. (Page #2, Line #10)

Q5: Page 2, Line 30: The unit of average rainfall should be 'cm/yr'.

Reply: Corrected. (Page #3, Line #9)

Q6: Authors should give a brief introduction about sources and types of aerosols in this site. And the size range of aerosol instrument, ranging from 5nm to 30 um, is very broad. Details about quality of surface instruments and calibration should be added.

Reply: The site is a small village in Western Ghats and tourists come during summer. Also, site is surrounded by thick vegetation with very high rainfall during summer monsoon season. Recent analysis with a chemical speciation monitor shows that sub-micron aerosols mainly consist of organics (77%), sulfates (14%), chlorides (4%). (Page #3, Line #15) Aerosol concentration and size distribution were measured using a Wide-Range Aerosol Spectrometer (WRAS) manufactured by GRIMM, Germany which is a combination of SMPS (Scanning Mobility Particle Sizer), measures particles in the size range 5 nm to 350 nm and APS (Aerosol Particle Sizer), measures particles in the size range from 350 nm to 32 $\mu$m. Combining SMPS and APS, the WRAS is capable of measuring the particle concentration covering full size range (5 nm –

32 $\mu$m) of atmospheric particles in 72 channels with a minimum scan time of 5 minutes. This is a stand-alone system with an automatic sample air dehumidification and condensate removal system in the condensation particle counter (CPC). SMPS was factory calibrated and APS calibration was done periodically with Polystyrene Latex (PSL) spheres of 300nm and measured by atomizing the hydrosol using a nebulizer; the resulting aerosol is dried and measured the size distribution of the aerosol particles. (The above detailed information is included in the revised manuscript Page #3, Line #25)

Q7: Define of $\varepsilon_{ij}\varepsilon_i$ in equation 6

Reply: $\varepsilon_i$ is a dimensionless parameter depends on the spectral shape of the cloud droplet size distribution. In the equation 6, $\varepsilon_{ij}\varepsilon_i$ is the intercept of the power law fit between $\varepsilon_i$ and L/N and the exponent $b\varepsilon_i$ gives the dispersion effect. (Page #6, Line #8)

Q8: For Figure 1 and 2, the LWC is fixed about 0.15g/m3, while in Figure 3 the LWC is fixed from 0.20g/m3 to 0.22g/m3. Why don't use the consistent LWC bin?

Reply: As reviewer suggested we have changed the figure 1a and 1b, with consistent LWC bins (0.20 – 0.23) as given in figure 3. (Page #14)

Q9: Page 8, Line 9: explain the calculation step from 0.07 to 21

Reply: According to Liu et.al, 2008, $3\times b\varepsilon_i$ in percentage is the dispersion offset ie, ($3 \times 0.07$). (Page #9, Line #4)

Q10: Cloud effective radius going down with aerosol concentration increasing might be due to the aerosol indirect effect as the author discussed in this article. It might also be due to the aerosol semi-direct effect (Huang et al., 2006a, 2006b, 2008 and 2010, 2014), which is very popular over the Asia region. Author should clarify this problem in the article.

Reply: Authors agree that semi-direct effect is dominant in Asia region during dry

season where in biomass burning and anthropogenic aerosols are dominant. As this study focuses only during monsoon (wet) season, we can see percentage/fraction of absorbing aerosol is very less compared to other aerosols. Organics and chlorides are dominant during this season and the semi-direct effect due to absorbing aerosol (eg., BC) is comparatively very less as compared to aerosol indirect effect. So the aerosol semi-direct effect is not addressed in this study. (Page #2, Line #23)

Please also note the supplement to this comment:
http://www.atmos-chem-phys-discuss.net/acp-2015-1057/acp-2015-1057-AC1-supplement.pdf

[Figure]

**Supplement:**

[revised manuscript text omitted]

**2.1. Wide range aerosol spectrometer (WRAS)**

Aerosol concentration and size distribution were measured using a Wide-Range Aerosol Spectrometer (WRAS) manufactured by GRIMM, Germany which is a combination of SMPS (Scanning Mobility Particle Sizer), measures particles in the size range 5 nm to 350 nm and APS (Aerosol Particle Sizer), measures particles in the size range from 350 nm to 32 μm. Combining SMPS and APS, the WRAS is capable of measuring the particle concentration covering full size range (5 nm – 32 μm) of atmospheric particles in 72 channels with a minimum

scan time of 5 minutes. This is a stand-alone system with an automatic sample air dehumidification and condensate removal system in the condensation particle counter (CPC).

SMPS was factory calibrated and APS calibration was done periodically with Polystyrene Latex (PSL) spheres of 300 nm and measured by atomizing the hydrosol using a nebulizer; the resulting aerosol is dried and measured the size distribution of the aerosol particles.

[revised manuscript text omitted]

---

## Referee Comment (RC2) · Anonymous Referee #2 · 25 Apr 2016

This paper describes a series of ground based measurements made from a hill top site in south western India during the summer monsoon season when the site was subjected to cloud for extended periods. The authors use both aerosol and cloud data to investigate the relationship between the aerosol indirect effect derived from cloud droplet number and also from effective radius and the effective dispersion. The authors report relationships and compare and contrast these with other previous publications. In addition, the authors go on to conclude that there is a significant difference between AIE derived from CDNC and Reff and this difference can be explained by the need to account for dispersion in the derivation of the former. My main concerns with the result are that there is no evidence presented to show the statistical robustness of

the correlations used to determine AIE and their susceptibility to biases, particularly by incorporating data at low CCN concentrations which appear fewer in number. In addition, there is no discussion of how the aerosol data were sampled (see detailed comments below). I would like to see these aspects of the papers developed before recommending publication in ACP.

Comments: I am very surprised that only 20 hours of data are selected for this analysis if, as the authors say, the site is covered by warm continental clouds most of the time during the summer monsoon season. It would be very useful to have statistics on the cloud frequency during the whole sample period and the method by which cloud events were screened for removal of precipitation. Without this it is impossible to gauge whether bias has been introduced into the sampling through data selection.

Page 2, line 10-13 and elsewhere: Given the paper is about the differences in AIE based on different methods of representing it the authors need to say how AIE was derived in each of the past work they cite.

Page 3: the aerosol instrumentation is not described, how were the number concentration measurements as a function of size between 5 nm and 30 um measured? How were the larger particles (>1um) sampled, was an inlet used and if so what was its transmission? If the large aerosols were measured at ambient humidities how were large aerosol separated from cloud droplets or were these counted as part of the same? Are the aerosol measured at the hill top in cloud, or below the cloud base? If the former, how are the interstitial aerosol collected, if the latter, how is flow connectivity established? This section needs a much more thorough description.

Figure 1a shows the relationship between CDNC and the total aerosol number and CCN. There is considerable curvature for low CCN and high aerosol load. This comes as little surprise since the aerosol concentration includes particle sizes from 5 nm upwards, which at times dominate the aerosol number concentrations but play no role in cloud activation. I fail to understand why figure 1a and 1b are plotted in the way

they are since the only comment in the paper is about the relationship between cloud droplet number and effective diameter which cannot be discerned from a single figure and instead the confusing use of total aerosol is included. It would be better to plot CDNC against Reff and colour the plot by CCN which would, I think, be far clearer to interpret.

Page 6, lines 1-3: Given the argument made here I cant help thinking that figure 2 would be better presented as a plot of ED (Reff) versus LWC and coloured by CDNC. That is the way the argument is made at least.

Page 6, lines 8 and 13: A statement is made that the correlations between CDNC and CCN are statistically significant, no such statement is made for the relationship between Reff and CCN, is the latter also statistically significant? I would like to see the method by which the significance of these relationships are tested statistically as by eye both figures 3a and 3b appear to have rather a low correlation, particularly figure 3b and I would take a little more convincing before I am persuaded that they are robust especially as the plots are on log-log axes. Given that the whole premise of the paper rests on the AIE being 30-40% different when derived from Nc than when derived from Reff there has to be a full uncertainty estimation of the slope I am particularly concerned that the plateau and tail off in Reff at low CCN concentrations greatly skews the slope of the fit. If data points below CCN=1000 cm-3 are excluded from fig 3b it would not surprise me if slopes of around -0.07 or steeper were derived which would be very close to the estimates determined by the CDNC data. A much more complete evaluation of the statistical robustness of the data and the possible biases at the ends of the data set is needed before I am willing to believe the differences the authors purport to show and conclude in lines 2026 of page 6. Given this LWC is at the maximum in the frequency distribution (figure 4) and shows the maximum difference between AIEn and AIEs then I suggest that a similar analysis is carried out for the different LWC bins to establish which parts, if any, of the distribution in figure 5 are robust statistically and may or may not be subject to the low CCN concentration biases shown in figure 3.

Page 8 lines 11-14: It is important to report here how the previous authors calculated the AIE from data as they are reporting a difference between AIEn and AIEs.

Minor Comments: Page 2, line 23: It would be good to be clear about the regional location and not just name the field site at this point. Page 3 line 8: A number of parameters are introduced without definition. For example CDNC here but earlier AIEn and AIEs

Page 3, lines 9-10: How were the non-rainy conditions defined?

Page 3, , lines 22-24: The CDP does not measure effective diameter, it measures cloud droplet number as a function of size, the other parameters are derived. The authors should be clear about how this was done.

Page 3, lines 29-30: It would be good to see a mathematical definition of how the spectral width of the droplet size distribution and the relative dispersion were calculated.

Page 5, lines 4-5: I do not believe the article show state that it "demonstrates" anything in a methods section. It is best to state what the paper seeks to achieve at this point in the text.

Page 5, lines 15-16: is the term b(beta) a percentage and is it an offset or an enhancement in the Twomey effect? Be clearer in the definition.

Minor Corrections: Page 1, Line 29: Twomey 1974

Page 2, line 1: "...but the field studies of the indirect aerosol effect shows..." should be show

Page 2, line 3: define epsilon, I realise this is defined on page 5 but it needs to be introduced as the relative dispersion at this point.

Page 2, line 6, "...cloud parcel(s) woith droplet(s) of the same..."

Page 2, line 8: show not shows

[Figure]

Page 2, line 9: "...and a slight decrease.."

Page 2, line 10: relationship(s)

Page 2, line 14: "...indicated that (the) dispersion effect..." Introduction: Throughout the introduction the authors mix up how they refer to their citations. At times this is done by reference to the work as a paper, eg (The paper by) Smith and Jones shows..., and at times by reference to the authors "...whereas Smith and Jones argue that. This needs to be consistent.

Page 2, line 20: "...decrease(s) the spectral width and in turn enhance(s)..."

Page 2, line 27: "(The recently set up..."

Page 2, line 29: "...situated in (the) Western Ghats.."

Page 2, line 31: "...during (the) summer monsoon..."

Page 3, line 1-2: "Interestingly, observations from the laboratory have shown that...."

Page 3, line 3-4: "...The aerosol and CCN concentration (measurements) shows that the region (experiences) higher aerosol concentration(s) during monsoon season..."

Page 3, line 17: "..in which (a) super saturated water vapour..."

Page 3, lines 15-20: include a reference to the CCN, typically Roberts and Nenes or similar.

Page 3, line 23-25: "...which is a combination of Cloud Droplet probe (CDP) [and a hotwire probe. The CDP measures the] cloud droplet size distribution and concentration from 3 to 50 $\mu$m, categorized 25 into one of 30 channels."

Page 3: define DSD

Page 4, line 19 and equation 3: It may be best to use n rather than N for the number of bins to clearly differentiate with $N_c$.

Page 4, line 19: "particle count"

Page 4, line 20: "….use (a) 1/3 power law.."

Page 4: in equation 4, the liquid water content is given as L yet elsewhere it is defined as LWC, it needs to be consistent.

Page 5, line 2: "…which is a function of (the) spectral shape of (the) cloud droplet size distribution.."

Page 5, line 3: "…for estimating AIE"

Page 5, line 4: "…this study (is) uniquely different…"

Page 5 line 6: "…to (a) Gamma distribution.."

Page 5, line 10: "..as the ratio of (the) standard…"

Page 5, line 13: "…to explain (the) dispersion effect."

Page 5, lines 15-16: "defined as the percentage of (the) offset/enhancement (in the) Twomey cooling effect due to the dispersion in the cloud droplet size distribution."

Page 5, equation (6): define alpha_beta

Page 5, line 21: why introduce ED without defining it as effective diameter except in figure 1 when you have already defined effective radius. I suggest redrawing figure 1 and also figure 5.

Page 6, line 12: "The linear fit to (the) log-log plot…"

Page 6, line 27: but the AIEs have already been estimated?

Page 7 line 11: "may cause (a) large number of"

Page 7, line 12: "and reduces the"

Page 7, line 17: "cloud albedo thus tend(ing) to reduce the AIE"

Page 8, line 4: "..spectra, has been calculated from CDP data...."

Page 8, lines 17-23: Why introduce DE as a term at this stage. Remove it.

Page 15: formatting of the figure caption needs correcting

———————————————————

---

## Author Response (AR1)

**Response to the Anonymous referee #2**

**Manuscript: Investigation of aerosol indirect effects on monsoon clouds using ground-based measurements over a high-altitude site in Western Ghats**
**by Anil Kumar et al.**

We would like to thank the Anonymous referee for his valuable suggestions which helped to improve the quality of this manuscript. Suggested corrections and modifications were included in the manuscript, which are highlighted with blue color.

**Comment 1**: I am very surprised that only 20 hours of data are selected for this analysis if, as the authors say, the site is covered by warm continental clouds most of the time during the summer monsoon season. It would be very useful to have statistics on the cloud frequency during the whole sample period and the method by which cloud events were screened for removal of precipitation. Without this it is impossible to gauge whether bias has been introduced into the sampling through data selection.

**Reply**: Cloud probe was not operated continuously, but during rain free cloudy conditions only. Cloud probe data during rainy /drizzle conditions were not considered in this analysis by utilizing collocated measurements of rain rate using an impact disdrometer. (Page 3, lines 24-25)

**Comment 2**: Page 2, line 10-13 and elsewhere: Given the paper is about the differences in AIE based on different methods of representing it the authors need to say how AIE was derived in each of the past work they cite.

**Reply**: Many of the past work used to derive aerosol number effect by either analyzing the changes of cloud droplet number concentration due to change in aerosol number concentration ($IE_1$), or the other by measuring the change in cloud effective radius with respect to aerosol number concentration ($IE_2$). Very few studies reported both $IE_1$ and $IE_2$, however detailed inferences why both estimates differ is not discussed. This is first such study discussing the

differences in AIE estimates between both the methods.    Similar results of higher $IE_1$ as compared to $IE_2$   was reported by Pandithurai et al. (2012) using the aircraft measurements over Indian region which used similar method. Usually with aircraft measurements sub-cloud aerosols are correlated with in-cloud parameters to derive AIE. However, in this study collocated surface measurements of aerosol and cloud parameters are used.

**Comment 3**: Page 3: the aerosol instrumentation is not described, how were the number concentration measurements as a function of size between 5 nm and 30 um measured? How were the larger particles (>1um) sampled, was an inlet used and if so what was its transmission? If the large aerosols were measured at ambient humidities how were large aerosol separated from cloud droplets or were these counted as part of the same? Are the aerosol measured at the hill top in cloud, or below the cloud base? If the former, how are the interstitial aerosol collected, if the latter, how is flow connectivity established? This section needs a much more thorough description.

**Reply**:

As suggested by the reviewer, the details of the aerosol instrument used in this study are given in the revised manuscript.

Aerosol concentration and size distribution were measured using a Wide-Range Aerosol Spectrometer (WRAS) manufactured by GRIMM, Germany which is a combination of SMPS (Scanning Mobility Particle Sizer), measures particles in the size range from 5 nm to 350 nm and APS (Aerosol Particle Sizer), measures particles in the size range from 250 nm to 32 μm.  Due to the **large particle size range** two different measurement principles have been used.   For particles between 0.25 μm (250 nm) and 32 μm (31 size channels ) an Optical Particle Counter (OPC – light scattering) is used, and a Scanning Mobility Particle Sizer + Counter (SMPS+C) consisting of a Differential Mobility Analyser (DMA) and a Condensation Particle Counter (CPC) is used for particle sizes from 5 up to 350 nm (44 size channels).  Data is reported in 71 size channels, as there are few common size channels for OPC and SMPS+C.

Air Sampling System comprising of Air Inlet, 1 meter common sampling pipe in SS with built-in **Nafion Dryer**, directly mounted on top of OPC/Aerosol Spectrometer. Particle losses are low

due to the special large diameter Nafion tubing used in the design.  This eliminates the need for heating the sample gas stream, preserving volatile particulate component in the sample.  Straight configuration minimizes the turbulent flow that can negatively affect measurements.  Dryer is continuously regenerating which eliminates the need to repeatedly replace the dessicant.

The aerosol measurements are taken at hill top where cloud base touches the observatory or fully covers the lab.  Some of the large aerosols measured at ambient humidity might be counted as cloud droplets but their number concentrations were very small.   For aerosol number concentration, mainly SMPS measurements range from 5 nm to 350 nm does not include large aerosols.   Due to these issues, not much aerosol measurements were considered in this manuscript.

**Comment 4**: Figure 1a shows the relationship between CDNC and the total aerosol number and CCN. There is considerable curvature for low CCN and high aerosol load. This comes as little surprise since the aerosol concentration includes particle sizes from 5 nm upwards, which at times dominate the aerosol number concentrations but play no role in cloud activation. I fail to understand why figure 1a and 1b are plotted in the way they are since the only comment in the paper is about the relationship between cloud droplet number and effective diameter which cannot be discerned from a single figure and instead the confusing use of total aerosol is included. It would be better to plot CDNC against Reff and colour the plot by CCN which would, I think, be far clearer to interpret.

**Reply**: Thanks.  As suggested and also to improve the clarity of the information, Figure 1 is revised to depict CDNC vs Reff for different CCN values.  This clearly illustrates higher CCN and CDNC results lower ED values and vice versa.

**Comment 5**: Page 6, lines 1-3: Given the argument made here I can't help thinking that figure 2 would be better presented as a plot of ED (Reff) versus LWC and coloured by CDNC. That is the way the argument is made at least.

**Reply**: Figure 2 is also re-plotted as suggested.

**Comment 6**: Page 6, lines 8 and 13: A statement is made that the correlations between CDNC and CCN are statistically significant, no such statement is made for the relationship between Reff and CCN, is the latter also statistically significant? I would like to see the method by which the significance of these relationships are tested statistically as by eye both figures 3a and 3b appear to have rather a low correlation, particularly figure 3b and I would take a little more convincing before I am persuaded that they are robust especially as the plots are on log-log axes. Given that the whole premise of the paper rests on the AIE being 30-40% different when derived from Nc than when derived from Reff there has to be a full uncertainty estimation of the slope I am particularly concerned that the plateau and tail off in Reff at low CCN concentrations greatly skews the slope of the fit. If data points below CCN=1000 cm$^{-3}$ are excluded from fig 3b it would not surprise me if slopes of around -0.07 or steeper were derived which would be very close to the estimates determined by the CDNC data. A much more complete evaluation of the statistical robustness of the data and the possible biases at the ends of the data set is needed before I am willing to believe the differences the authors purport to show and conclude in lines 20-26 of page 6. Given this LWC is at the maximum in the frequency distribution (figure 4) and shows the maximum difference between AIEn and AIEs then I suggest that a similar analysis is carried out for the different LWC bins to establish which parts, if any, of the distribution in figure 5 are robust statistically and may or may not be subject to the low CCN concentration biases shown in figure 3.

**Reply**: The correlation between CCN & CDNC and CCN & Reff are statistically significant for all LWC bins considered here.  The significant test (Students T-test) for CCN Vs CDNC and CCN Vs Reff for different LWC bins showed that the correlation coefficient R is consistently high for all of the LWC bins < 0.45g/m3, and they are statistically significant with strong confidence level.

[Figure]

As suggested by the reviewer, data points below CCN=1000 #/cm3 is excluded and found the slope of CCN Vs CDNC and CCN Vs Reff, the new plot is given below. Still higher $AIE_n$ values can be noted as compared to $AIE_s$. The difference in $AIE_n$ and $AIE_s$ can be minimized by applying the DE to $AIE_n$. For example at 0.2g/m3 LWC bin $AIE_n$ is 0.0967 and $AIE_s$ is 0.07. The dispersion offset of 25.8% (0.0249) is reduced from 0.967 gives 0.071, which is approximately same as $AIE_s$. After excluding data points below CCN=1000#/cm3 there is a slight increase in correlation, which can be noted in all LWC bins.

(Page 6, line 25-26 - Page 7, lines 1)

[Figure]

[Figure]

**Comment 7**: Page 8 lines 11-14: It is important to report here how the previous authors calculated the AIE from data as they are reporting a difference between AIEn and AIEs.

**Reply**: In most of the previous works, the AIE is calculated mainly by one of the methods either aerosol vs cloud droplet number or aeroso vs $R_{eff}$. Few studies reported accurate representation of AIE needs to consider i) dispersion effect, ii) entrainment effects. In this study, we present both the methods and systematic difference which can be minimized by considering the dispersion effect. (Page 9, lines 11-15)

**Minor Comments**

Comment 8: Page 2, line 23: It would be good to be clear about the regional location and not just name the field site at this point.

Reply: Necessary changes were made. (Page 3, line 17)

Page 3 line 8: A number of parameters are introduced without definition. For example CDNC here but earlier AIEn and AIEs.

Reply: Necessary changes were made. (Page 3, line 21)

Comment 9: Page 3, lines 9-10: How were the non-rainy conditions defined?

Reply: Data was taken during non rainy condition. Disdrometer data is also used to identify the non rainy conditions. (Page 3, lines 24-25)

Comment 10: Page 3, lines 22-24: The CDP does not measure effective diameter, it measures cloud droplet number as a function of size, the other parameters are derived. The authors should be clear about how this was done.

Reply: The CDP measures the cloud droplet size distribution (DSD) and concentration of droplet sizes from 3 to 50 μm, categorized into 30 channels. Effective diameter is derived and the details are included in the revised manuscript. (Page 4, lines 5-6)

Comment 11: Page 3, lines 29-30: It would be good to see a mathematical definition of how the spectral width of the droplet size distribution and the relative dispersion were calculated.

Reply: As suggested by the reviewer, mathematical definitions of $r_m$, $\sigma$ and $\varepsilon$ are included in the revised manuscript. (Page 4, line 12)

Comment 12: Page 5, lines 4-5: I do not believe the article show state that it "demonstrates" anything in a methods section. It is best to state what the paper seeks to achieve at this point in the text.

Reply: The sentence was not appropriate in the methodology section, so it is removed.

Comment 13: Page 5, lines 15-16: is the term b(beta) a percentage and is it an offset or an enhancement in the Twomey effect? Be clearer in the definition.

Reply : $b_\beta>0$ indicates the dispersion effect offsets the Twomey effect. The definition is changed in the manuscript. (Page 6, line 11)

Minor Corrections:

Page 1, Line 29: Twomey 1974

Corrected (Page 1, line 29)

Page 2, line 1: ": : :but the field studies of the indirect aerosol effect shows: : :" should be show

Corrected (Page 2, line 2)

Page 2, line 3: define epsilon, I realise this is defined on page 5 but it needs to be introduced as the relative dispersion at this point.

Thanks. Corrected (Page 2, line 3)

Page 2, line 6, ": : :cloud parcel(s) woith droplet(s) of the same: : :"

Corrected (Page 2, line 6)

Page 2, line 8: show not shows

Corrected (Page 2, line 9)

Page 2, line 9: ": : :and a slight decrease.."

Corrected (Page 2, line 9)

Page 2, line 10: relationship(s)

Corrected (Page 2,line 11)

Page 2, line 14: ": : :indicated that (the) dispersion effect: : :"

Corrected (Page 2, line 5)

Introduction: Throughout the introduction the authors mix up how they refer to their citations. At times this is done by reference to the work as a paper, eg (The paper by) Smith and Jones shows: : :, and at times by reference to the authors ": : :whereas Smith and Jones argue that. This needs to be consistent.

Corrected

Page 2, line 20: ": : :decrease(s) the spectral width and in turn enhance(s): : :"

Corrected (Page 2, lines 20-21)

Page 2, line 27: "(The recently set up: : :"

Corrected (Page 3, line 6)

Page 2, line 29: ": : :situated in (the) Western Ghats.."

Corrected (Page 3, line 8)

Page 2, line 31: ": : :during (the) summer monsoon: : :"

Corrected (Page 3, line 10)

Page 3, line 1-2: "Interestingly, observations from the laboratory have shown that: : :."

Corrected (Page 3, line 11)

Page 3, line 3-4: ": : :The aerosol and CCN concentration (measurements) shows that the region (experiences) higher aerosol concentration(s) during monsoon season: : :"

Corrected (Page 3, lines 13-14)

Page 3, line 17: "..in which (a) super saturated water vapour: : :"

Corrected (Page 3, line 31)

Page 3, lines 15-20: include a reference to the CCN, typically Roberts and Nenes or similar.

Thanks.  Incuded (Page 3, line 28)

Page 3, line 23-25: ": : :which is a combination of Cloud Droplet probe (CDP) [and a hotwire probe. The CDP measures the] cloud droplet size distribution and concentration from 3 to 5um, categorized into one of 30 channels."

Corrected (Page 4, lines 5-6)

Page 3: define DSD

Defined (Page 4, line 6)

Page 4, line 19 and equation 3: It may be best to use n rather than N for the number of bins to clearly differentiate with Nc.

Corrected (Page 5, line 14)

Page 4, line 19: "particle count"

Corrected (Page 5, line 14)

Page 4, line 20: ": : :.use (a) 1/3 power law.."

Corrected (Page 5, line 15)

Page 4: in equation 4, the liquid water content is given as L yet elsewhere it is defined as LWC, it needs to be consistent.

Corrected

Page 5, line 2: ": : :which is a function of (the) spectral shape of (the) cloud droplet size distribution.."

Corrected (Page 5, line 22)

Page 5, line 3: ": : :for estimating AIE"

Corrected (Page 5, line 23)

Page 5, line 4: ": : :this study (is) uniquely different: : :"

Corrected

Page 5 line 6: ": : :to (a) Gamma distribution.."

Corrected (Page 6, line 1)

Page 5, line 10: "..as the ratio of (the) standard: : :"

Corrected (Page 6, line 5)

Page 5, line 13: ": : :to explain (the) dispersion effect."

Corrected (Page 6, line 8)

Page 5, lines 15-16: "defined as the percentage of (the) offset/enhancement (in the) Twomey cooling effect due to the dispersion in the cloud droplet size distribution."

Corrected (Page 6, line 11)

Page 5, equation (6): define alpha_beta

Defined in the revised manuscript. (Page 6, line 10)

Page 5, line 21: why introduce ED without defining it as effective diameter except in figure 1 when you have already defined effective radius. I suggest redrawing figure 1 and also figure 5.

Corrected, redrawn figure 1 and figure 2 as suggested.

Page 6, line 12: "The linear fit to (the) log-log plot: : :"

Corrected (Page 7, line 7)

Page 6, line 27: but the AIEs have already been estimated?

Corrected (Page 7, line 22)

Page 7 line 11: "may cause (a) large number of"

Corrected (Page 8, line 7)

Page 7, line 12: "and reduces the"

Corrected (Page 8, line 8)

Page 7, line 17: "cloud albedo thus tend(ing) to reduce the AIE"

Corrected (Page 8, line 14)

Page 8, line 4: "..spectra, has been calculated from CDP data: : :."

Corrected (Page 8, line 29)

Page 8, lines 17-23: Why introduce DE as a term at this stage. Remove it.

Corrected (Page 9, lines 18-21)

Page 15: formatting of the figure caption needs correcting

Corrected.